# SUMOylation contributes to proteostasis of the chloroplast protein import receptor TOC159 during early development

**Sonia Accossato, Felix Kessler*, Venkatasalam Shanmugabalaji***

Laboratory of Plant Physiology, University of Neuchâtel, Neuchâtel, Switzerland

**Abstract** Chloroplast biogenesis describes the transition of non-photosynthetic proplastids to photosynthetically active chloroplasts in the cells of germinating seeds. Chloroplast biogenesis requires the import of thousands of nuclear-encoded preproteins by essential import receptor TOC159. We demonstrate that the small ubiquitin-related modifier (SUMO) pathway crosstalks with the ubiquitin–proteasome pathway to affect TOC159 stability during early plant development. We identified a SUMO3-interacting motif (SIM) in the TOC159 GTPase domain and a SUMO3 covalent SUMOylation site in the membrane domain. A single K to R substitution (K1370R) in the M-domain disables SUMOylation. Compared to wild-type TOC159, TOC159K1370R was destabilized under UPS-inducing stress conditions. However, TOC159K1370R recovered to same protein level as wild-type TOC159 in the presence of a proteasome inhibitor. Thus, SUMOylation partially stabilizes TOC159 against UPS-dependent degradation under stress conditions. Our data contribute to the evolving model of tightly controlled proteostasis of the TOC159 import receptor during proplastid to chloroplast transition.

***For correspondence:**
felix.kessler@unine.ch (FK);
shanmugabalaji.venkatasalam@
unine.ch (VS)

**Competing interests:** The authors declare that no competing interests exist.

## Introduction

Chloroplasts are unique organelles that carry out photosynthesis. Although chloroplasts contain their own genome, the majority of chloroplast proteins are encoded by the nuclear genome and synthesized as preproteins in cytosol, and these preproteins are imported into the chloroplast through TOC-TIC complexes (**T**ranslocon at the **O**uter or **I**nner membrane of the **C**hloroplast) (*Jarvis and López-Juez, 2013*). The core of the TOC complex contains two related GTP-dependent preprotein receptor GTPases, TOC159 and TOC33, which interact with a β-barrel membrane protein, TOC75, that forms a protein-conducting channel, and is regulated by specific interactions with nuclear-encoded preproteins (*Schnell et al., 1994*; *Kessler et al., 1994*; *Jarvis et al., 1998*). TOC159 is a major point of entry for highly abundant photosynthesis-associated preproteins arriving at the translocon complex. It is therefore regarded as the major chloroplast protein import receptor. TOC159 has a three-domain structure: a highly acidic N-terminal domain (A-domain), a central GTP-binding domain (G-domain), and a C-terminal membrane anchor domain (M-domain). Chloroplast biogenesis, the transition of a non-photosynthetic proplastid to a photosynthetically active chloroplast, depends on the essential import receptor TOC159 and its mutation results in non-photosynthetic albino plants (*Bauer et al., 2000*).

Recent studies have shown that during the etioplast to chloroplast transition, the TOC components are ubiquitinated by a novel chloroplast RING-type E3 ubiquitin ligase SP1 (suppressor of ppi1 locus 1). Ubiquitinated TOC components are extracted from the chloroplast outer envelope membrane with the help of SP2 (an Omp85-like β-barrel protein) and Cdc48 (a cytosolic AAA+ chaperone) providing the extraction force. The ubiquitinated TOC component is degraded by the 26S proteasome in the cytosol (*Ling et al., 2012*; *Ling et al., 2019*). This proteolytic pathway has been named CHLORAD. In a different context, chloroplast biogenesis takes place during the proplastid to

chloroplast transition. It is dependent on the plant hormone gibberellic acid (GA). Under unfavorable seed germination conditions when gibberellic concentrations are low, a DELLA (RGL2) (a negative regulator of GA signaling) promotes the ubiquitylation and degradation of TOC159 by the 26S proteasome to insertion into the outer membrane of the chloroplast. This mechanism delays the onset of chloroplast biogenesis at an early developmental stage and has been shown to be independent of SP1 and presumably CHLORAD (*Shanmugabalaji et al., 2018*).

A recent study employed a yeast-two-hybrid screen to identify putative Arabidopsis small ubiquitin-like modifier (SUMO) substrates. The E2 SUMO conjugating enzyme (SCE1) was used as bait, and TOC159 was identified as a high-probability interaction candidate. TOC159 contains a predicted SUMO attachment site (ψKXE/D) (*Elrouby and Coupland, 2010*). The SUMO, a 11 kDa protein, covalently modifies a large number of proteins. SUMO-dependent regulation is involved in many cellular processes, including gene expression, signal transduction, genome maintenance, protein localization, and activity (*Gill, 2004*). SUMOylation plays a crucial role in plant development and stress responses (*Elrouby, 2017*; *Augustine and Vierstra, 2018*; *Morrell and Sadanandom, 2019*). This reversible and dynamic SUMOylation starts with the attachment of SUMO to the target protein by a conjugation pathway, mechanistically analogous to the ubiquitylation system (*Augustine and Vierstra, 2018*; *Novatchkova et al., 2012*). The target protein covalently modified by SUMO performs specific functions, which may subsequently be reversed by SUMO proteases that hydrolyze the isopeptide bond between SUMO and the target protein (*Yates et al., 2016*). In addition, SUMO can also interact with target proteins through a SUMO-interacting motif (SIM). The non-covalent SUMO–SIM interaction may work as a molecular signal for protein–protein interaction and affect stability of proteins (*Geiss-Friedlander and Melchior, 2007*).

In this study, we address the role of SUMO modification of TOC159 in the context of the proplastid to chloroplast transition and investigate both covalent SUMOylation and non-covalent SUMO interaction. The biochemical and genetic evidence showed that the TOC159 G-domain contains a SIM that interacts with the SUMO3 and that SUMOylation of TOC159 at the M-domain is the key regulatory mechanism to protect it from further depletion under low GA conditions. The data provides new insight for TOC159 SUMO-binding and SUMOylation, demonstrating that SUMOylation positively influences protein stability with regard to the UPS. Thereby SUMOylation contributes to the proteostatic fine-tuning of TOC159 levels during TOC159-dependent chloroplast biogenesis in early plant development.

## Results

### SUMO3 interacts with the TOC159 G-domain and is SUMOylated at the TOC159 M-domain

An earlier study using an in vitro SUMOylation assay showed that of the three SUMO isoforms (SUMO1, SUMO2, and SUMO3), only SUMO3 SUMOylated TOC159 (*Elrouby and Coupland, 2010*). TOC159 has a N-terminal A- (acidic-), a central G- (GTP-binding domain), and a C-terminal M- (membrane) domain (*Bauer et al., 2000*). The A-domain is known to be non-essential for TOC159 function and exquisitely sensitive to protease activity (*Agne et al., 2009*; *Agne et al., 2010*). It was excluded from our DNA constructs and only the TOC159 G- and M-domains (TOC159GM) were used. In addition to a covalent SUMOylation site, TOC159 may also have SIM. To investigate the possibility of SUMO interaction with TOC159GM, we used the GPS-SUMO prediction algorithm (http://sumosp.biocuckoo.org/online.php) to search for SIM in TOC159GM (*Zhao et al., 2014*). Based on the search results, there is a predicted SIM (VKVLP) in the G-domain (*Figure 1A*). To analyze the physical interaction between TOC159GM and SUMO isoforms, we performed yeast two-hybrid assays using TOC159G and TOC159M separately as baits. TOC159G interacted exclusively with SUMO3, but not SUMO1 and SUMO2. None of the SUMO isoforms interacted with TOC159M in the yeast two-hybrid assay (*Figure 1B*, *Figure 1—figure supplement 1*). We further confirmed the TOC159GM–SUMO3 interaction by co-immunoprecipitation of GFP-TOC159GM and SUMO3-MYC by in planta transient co-expression using the *Nicotiana benthamiana* system (*Figure 1C*).

We used the GPS-SUMO algorithm (http://sumosp.biocuckoo.org/online.php) to search for covalent SUMOylation sites in TOC159GM. A high scoring consensus SUMOylation site with a strongly conserved motif (TGVKLED) and containing a potentially SUMOylatable lysine (K1370) was identified

**Figure 1.** Small ubiquitin-related modifier (SUMO) interaction and SUMOylation of TOC159GM. (**A**) Schematic representation of TOC159GM indicating the predicted SUMO-interacting motif (SIM) in the G-domain. (**B**) Yeast two-hybrid interaction assay of TOC159 G with SUMO proteins on –Leu, –Trp and –Leu, –Trp, –His medium. AD, activation domain; BD, binding domain. (**C**) Transient expression of SUMO3-MYC, GFP-TOC159GM, and the combination of both in *Nicotiana benthamiana* leaves. Total protein extracts were subjected to immunoprecipitation with anti-GFP beads. The

*Figure 1 continued*

immunoprecipitated proteins from the expression of SUMO3-MYC (lane 1) and GFP-TOC159GM (lane 2) alone and the co-expression both (lane 3) were analyzed by western blotting using anti-GFP and anti-MYC antibodies. (D) Schematic representation of TOC159GM with indication of the predicted SUMOylation site K1370 (Lysine) at the M-domain. (E) Alignment of the conserved predicted K1370 SUMOylation sites in the M-domain of multiple species: *Arabidopsis thaliana* (At), *Pisum sativum* (Ps), *Solanum lycopersicum* (Sl), *Oryza sativa* (Os), and *Sorghum bicolor* (Sb) by using CLUSTAL Omega (1.2.4) multiple sequence alignment tool. (F) Transient expression of GFP-TOC159GM and GFP-TOC159GM-K/R (SUMO mutant, K1370 replaced with R) with and without SUMO3-MYC in *N. benthamiana* leaves. Total protein extracts were subjected to immunoprecipitation with anti-GFP beads. The immunoprecipitated proteins from the expression of GFP-TOC159GM (lane 1) and GFP-TOC159GM-K/R (lane 2) alone as well as the co-expression with SUMO3 (lanes 3 and 4) were analyzed by western blotting using anti-GFP, anti-MYC and anti-SUMO3 antibodies.

The online version of this article includes the following source data and figure supplement(s) for figure 1:

**Source data 1.** Source data for *Figure 1E*.
**Figure supplement 1.** Yeast two-hybrid interaction assay of TOC159 M-domain with SUMO proteins on –Leu, –Trp and –Leu, –Trp, and –His medium.
**Figure supplement 2.** Predicted SUMOylation sites at TOC159GM and in planta SUMOylation assay.

within the M-domain (*Figure 1D*, *Figure 1—figure supplement 2A*). The SUMOylation motif as well as K1370 of Arabidopsis are well conserved in other plants species (*Figure 1E*). To investigate the SUMOylation of TOC159GM, we selected the SUMO3 isoform based on the earlier in vitro study (*Elrouby and Coupland, 2010*). We infiltrated *Nicotiana benthamiana* with 35S-GFP-TOC159GM or GFP-TOC159GM-K/R (replacing lysine with a non-sumoylatable arginine residue at position 1370) each together with or without 35S-SUMO3-MYC. To analyze the infiltration experiments identical amounts of total extracts were subjected to immunoprecipitation using anti-GFP-beads followed by western blotting. An anti-GFP antibody was used to indicate total expression of GFP-TOC159GM and GFP-TOC159GM-K/R and resulted in bands of similar intensities in all four experiments. Anti-MYC and anti-SUMO3 were used to analyze conjugation of SUMO3-MYC to GFP-TOC159GM and GFP-TOC159GM-K/R. The western blotting using anti-MYC and anti-SUMO3 antibodies resulted in strong signals for GFP-TOC159GM (*Figure 1F*, lane 3) but only very weak ones for GFP-TOC159GM-K/R (*Figure 1F*, lane 4) when co-expressed with SUMO3-MYC. No signals were detected when the GFP-TOC159GM constructs were expressed in the absence of SUMO3-MYC (*Figure 1F*, lanes 1 and 2). Note that GFP-TOC159GM co-infiltration with SUMO3 resulted in a much higher molecular mass band when analyzed with anti-MYC or -SUMO3 than the main, strong GFP-TOC159GM band detected with anti-GFP. It therefore appears that upon co-expression with SUMO3-MYC only a small fraction of GFP-TOC159GM was present in the SUMOylated form and that it had a higher molecular mass than GFP-TOC159GM alone (*Figure 1F*). We further checked the SUMOylation assay by co-infiltration of GFP alone (control), GFP-TOC159GM, and GFP-TOC159GM-K/R together with SUMO3 in the *N. benthamiana* system. The co-immunoprecipitation gave strong signals for GFP-TOC159GM (*Figure 1—figure supplement 2B*, lane 2) but only very weak signals for GFP-TOC159GM-K/R (*Figure 1—figure supplement 2B*, lane 3) and no detectable signals for GFP alone (*Figure 1—figure supplement 2B*, lane 1) when using the anti-MYC antibody. The results confirmed specificity of TOC159GM SUMOylation.

## The non-SUMOylatable TOC159GM-K/R mutant complements the *ppi2* mutation

It has been demonstrated that the A-domain of TOC159 is dispensable but that the M-domain of TOC159 is essential for protein import into the chloroplast (*Lee et al., 2003*). Furthermore, it was demonstrated that TOC159GM alone without the A-domain could complement the albino *ppi2* phenotype (*Agne et al., 2009*). To characterize the effect of the K1370R mutation on the M-domain in vivo, we engineered transgenic lines expressing GFP-TOC159GM as well as GFP-TOC159GM-K/R under the TOC159 promoter in the *ppi2* background. Two independent transgenic lines of pTOC159-GFP-TOC159GM:*ppi2* and pTOC159-GFP-TOC159GM-K/R:*ppi2* (called GFP-TOC159GM:*ppi2* and GFP-TOC159GM-K/R:*ppi2* plants hereafter) were isolated and the genotypes were confirmed by PCR using specific primer pairs (*Figure 2A*, *Figure 2—figure supplement 1A*).

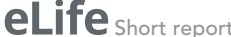

**Figure 2.** Complementation of *ppi2* (*toc159* mutant) by non-SUMOylatable TOC159GM-K/R. (**A**) Phenotypes of 7 days old seedling of *ppi2*, WT (Ws), GFP-TOC159GM:*ppi2*, and GFP-TOC159GM-K/R:*ppi2*. (**B**) Western blot analysis of total protein extracts from 7 days old seedlings of WT (Ws), GFP-TOC159GM:*ppi2* and GFP-TOC159GM-K/R:*ppi2*. The blot was probed with anti-GFP, -TOC75, and TOC33 antibodies. Actin was used as a loading control. (**C**) Chlorophyll levels of wild type, GFP-TOC159GM:*ppi2*, and GFP-TOC159GM-K/R:*ppi2* from 7 days old seedlings. Error bars indicate ± SEM

*Figure 2 continued on next page*

*Figure 2 continued*

(n = 4). (**D**) Confocal microscopy analysis of 7 days old GFP-TOC159GM:*ppi2* and GFP-TOC159GM-K/R:*ppi2* seedlings. Green fluorescence (GFP, left-hand panel), red fluorescence (chlorophyll, middle pane), and the overlay of the two (right-hand panel) are shown.

The online version of this article includes the following source data and figure supplement(s) for figure 2:

**Source data 1.** Source data for *Figure 2C*.
**Figure supplement 1.** Screening of *ppi2* plants for complementation with GFP-TOC159GM and GFP-TOC159GM-K/R.

Homozygous GFP-TOC159GM:*ppi2* and GFP-TOC159GM-K/R:*ppi2* transgenic lines gave green plants with almost identical chlorophyll concentrations that were very close to those of the wild type (*Figure 2C*). This indicated complementation of the albino *ppi2* phenotype. Western blotting analysis using GFP antibodies specifically detected GFP-TOC159GM and GFP-TOC159GM-K/R proteins. The blots revealed that TOC159GM and GFP-TOC159GM-K/R proteins accumulated to very similar levels in the respective transgenic lines. The blots were also probed with antibodies against TOC75 and TOC33, showing no significant differences between the two transgenic lines (*Figure 2B*, *Figure 2—figure supplement 1B,C*). To localize the GFP-TOC159GM and GFP-TOC159GM-K/R in vivo, 7 days old seedlings were analyzed by confocal fluorescence microscopy. Both GFP-TOC159GM and GFP-TOC159GM-K/R gave clear fluorescent signals at the chloroplast periphery that are consistent with outer envelope membrane localization (*Figure 2D*).

## TOC159GM-K/R accumulation is diminished when compared to TOC159GM under low GA conditions

Addition of paclobutrazol (PAC) to the Murashige–Skoog (MS) growth medium can be used to inhibit GA biosynthesis and results in low GA conditions (*Piskurewicz et al., 2008*). Under low GA during early germination, the DELLA (RGL2) destabilizes TOC159 via the ubiquitin–proteasome system (*Shanmugabalaji et al., 2018*). To explore a possible role of SUMOylation of K1370 on protein stability, GFP-TOC159GM:*ppi2* and GFP-TOC159GM-K/R:*ppi2* were allowed to germinate in the presence or absence of PAC. As reported earlier, the GFP-TOC159GM protein level was severely reduced in GFP-TOC159GM:*ppi2* seeds under low GA. In comparison, the level of GFP-TOC159-K/R in GFP-TOC159GM-K/R:*ppi2* seeds was even more diminished. Moreover, TOC75 and TOC33 levels were also lower in mutant TOC159GM-K/R:*ppi2* than in wild-type GFP-TOC159GM:*ppi2* under low GA, whereas their levels were the same in the untreated lines (*Figure 3A,B, Figure 3—figure supplements 1* and *2*). PAC-treated seeds, low in GA, accumulated very high levels of the RGL2 protein (*Piskurewicz et al., 2008*). We also compared RGL2 protein levels between TOC159GM:*ppi2* and TOC159GM-K/R:*ppi2* in the presence of PAC. The results revealed that there was no difference in RGL2 accumulation between the two lines (*Figure 3C, Figure 3—figure supplement 3A,B*).

## SUMOylation partially stabilizes the TOC159 under low GA conditions

To determine whether the diminished stability of GFP-TOC159GM-K/R under low GA in the presence of PAC is also due to UPS-mediated degradation, GFP-TOC159GM-K/R:*ppi2* and GFP-TOC159GM:*ppi2* seeds were germinated in the presence of PAC and subjected to treatment with or without the proteasome inhibitor MG132. Western blot analysis demonstrated that both GFP-TOC159GM-K/R and GFP-TOC159GM were rescued by MG132 and accumulated to the same level (*Figure 3E,F, Figure 3—figure supplement 4*). The results suggest that SUMOylation partially protects GFP-TOC159GM against UPS-mediated degradation under low GA when compared to GFP-TOC159GM-K/R.

## Accumulation of photosynthesis-associated proteins trended lower in TOC159GM-K/R:*ppi2* than TOC159GM:*ppi2*

The results so far demonstrated that non-SUMOylatable TOC159GM-K/R is significantly more susceptible to UPS-mediated degradation than wild-type TOC159GM under low GA. To address whether this has an effect on the accumulation of photosynthesis-associated proteins (the presumed

**Figure 3.** SUMOylation partially protects TOC159 from UPS-mediated degradation. (**A**) Immunoblotting of total protein extracts from 3 days old seedling of GFP-TOC159GM:*ppi2* and GFP-TOC159GM-K/R:*ppi2* grown in the presence or absence of paclobutrazol (PAC) (5 µM). WT (Ws) was used as the control for antibody specificity. The blot was probed with anti-GFP, -TOC75, and -TOC33 antibodies. Anti-actin was used for a loading control. (**B**) Specific bands corresponding to GFP, TOC75, TOC33, and actin were quantified from three independent experiments (**A**). The quantified bands

*Figure 3 continued on next page*

*Figure 3 continued*

were normalized to GFP-TOC159GM in GFP-TOC159GM:*ppi2* plants grown in the absence of PAC. Error bars indicate ± SEM (n = 3). two-tailed t test; *p<0.05; ***p<0.005. (C) Immunoblotting of total protein extracts from 3 days old GFP-TOC159GM:*ppi2* and GFP-TOC159GM-K/R:*ppi2* seedlings grown in the presence or absence of PAC (5 μM). The blot was probed with anti-GFP and -RGL2 antibodies. Anti-actin was used for a loading control. (D) Total protein extracts of 3 days old GFP-TOC159GM:*ppi2* or GFP-TOC159GM-K/R:*ppi2* grown seedling grown on PAC and subsequently treated with or without MG132 were analyzed by immunoblotting using anti-GFP antibodies and anti-UGPase for a loading control. (E) The specific bands corresponding to GFP and UGPase were quantified from three independent experiments (C). The quantified bands were normalized to GFP-TOC159GM in GFP-TOC159GM:*ppi2* without MG132. Error bars indicate ± SEM (n = 3). two-tailed t test; **p<0.01.

The online version of this article includes the following source data and figure supplement(s) for figure 3:

**Source data 1.** Source data for *Figure 3B, E*, *Figure 3—figure supplements 3* and *5*.

**Figure supplement 1.** Immunoblotting of total protein extracts from two independent GFP-TOC159GM:*ppi2* lines (3.1 and 3.12) and two independent GFP-TOC159GM-K/R:*ppi2* lines (4.2 and 4.8).

**Figure supplement 2.** Immunoblotting of total protein extracts from GFP-TOC159GM:*ppi2* and GFP-TOC159GM-K/R:*ppi2* grown in the presence or absence of PAC (5 μM).

**Figure supplement 3.** RGL2 protein accumulation in GFP-TOC159GM:ppi2and GFP-TOC159GM-K/R:ppi2under low GA conditions.

**Figure supplement 4.** Total protein extracts of 3 days old GFP-TOC159GM:*ppi2* or GFP-TOC159GM-K/R:*ppi2* grown in the presence of PAC and further treated with or without MG132, analyzed by immunoblotting using antibodies against GFP.

**Figure supplement 5.** The TOC159 SUMOylation-deficient line TOC159GM-K/R:*ppi2* accumulates photosynthesis-associated proteins trended lower compared to TOC159GM:*ppi2* under low GA conditions.

TOC159 cargo proteins), western blotting was carried out comparing GFP-TOC159GM-K/R:*ppi2* to GFP-TOC159GM:*ppi2*. It has been shown that the expression of most photosynthesis-associated genes did not change significantly in the presence of moderate and low PAC concentrations (5 and 1 μM) (*De Giorgi et al., 2015*). To eliminate the possibility of gene expression effects on protein accumulation, GFP-TOC159GM:*ppi2* and GFP-TOC159GM-K/R:*ppi2* seedlings were grown in the presence or absence of an intermediate 2 μM PAC concentration. The western blot analysis revealed that the accumulation of photosynthesis-associated proteins (PSBO1, PSBA, RBCL, RBCS, PETC, PSAD, ATPC, and LHCB2) systematically trended lower but not to statistically significant extent in TOC159GM-K/R:*ppi2* when compared to TOC159GM:*ppi2* (*Figure 3—figure supplement 5*).

## Discussion

It has been demonstrated previously that TOC159 physically interacts with the SUMO E2 enzyme in a yeast two-hybrid screen and that the SUMO3 isoform covalently SUMOylated TOC159 in an in vitro assay (*Elrouby and Coupland, 2010*). Apart from a covalent SUMOylation motif, the GPS-SUMO algorithm identified a conserved non-covalent SIM ('VKVLP') in the G-domain of TOC159. We confirmed this prediction using a yeast two-hybrid assay and co-immunoprecipitation experiment. It also revealed that the G-domain specifically interacted with the SUMO3 isoform and not with SUMO1 and 2. (*Figure 1B*). It has been shown in Arabidopsis that the GA receptor GID1 interacted with SUMO1 through a SIM that prevented its interaction with a DELLA protein (*Conti et al., 2014*). It is tempting to hypothesize that non-covalent binding of TOC159(SIM) to SUMO3 may modify interactions with SUMOylated complex partners such as TOC33/TOC75 or the DELLA (RGL2) under low GA conditions as these proteins also have predicted SUMOylation sites (http://sumosp.bio-cuckoo.org/online.php).

TOC159 has highly conserved SUMOylation motif ('TGV**K**LED') with a lysine at position 1370 (K1370) within the M-domain of TOC159. We established SUMOylation at K1370 using an in planta SUMOylation assay in *N. benthamiana* (*Figure 1F, Figure 1—figure supplement 2B*). Expression of non-SUMOylatable GFP-TOC159GM-K/R (containing the K1370R substitution) restored a wild-type green phenotype in the GFP-TOC159GM-K/R:*ppi2* Arabidopsis plants. Note that *ppi2* mutant plants normally have an albino phenotype. Based on the presence of wild-type levels of TOC75 and TOC33, it appeared that the TOC complex assembled normally in the GFP-TOC159GM-K/R:*ppi2* plants (*Figure 2A,B*). Confocal laser microscopy localized GFP-TOC159GM-K/R at the envelope of

the chloroplast (*Figure 2D*). Taken together, these results indicate that outer membrane insertion, TOC complex assembly, as well as chloroplast biogenesis take place normally under standard growth conditions despite the K1370R mutation and disabled SUMOylation.

In Arabidopsis, SUMOylation is triggered by environmental stimuli including biotic and abiotic stress. Hormone signaling (GA, auxin, brassinosteroids, and ABA) and development (root, seed, embryo, and meristem) are the main biological processes associated with SUMOylation in plants (*Conti et al., 2014*; *Miura et al., 2009*; *Ishida et al., 2009*; *Augustine et al., 2016*; *Orosa-Puente et al., 2018*; *Kwak et al., 2019*; *Srivastava et al., 2020*). While SUMOylation is independent of the ubiquitination pathway, there is the complex crosstalk between these two pathways. SUMOylation can promote the UPS-dependent degradation through SUMO-targeted Ub ligases (*Perry et al., 2008*). Contrary to this, SUMOylation may also protect from UPS-dependent degradation by blocking the ubiquitination of lysine residues (*Creton and Jentsch, 2010*; *Jentsch and Psakhye, 2013*; *Ramachandran et al., 2015*; *Rott et al., 2017*). Here, we explored the connection between TOC159 SUMOylation and the UPS-mediated TOC159 degradation under low GA during early developmental stages in seed germination.

We previously demonstrated that low GA concentrations brought about by PAC promote the DELLA (RGL2)-dependent TOC159 degradation via the UPS (*Shanmugabalaji et al., 2018*). Under low GA, GFP-TOC159GM-K/R accumulated to significantly (<50%) lower levels than wild-type GFP-TOC159GM in the respective overexpression lines. In addition, the TOC159-interacting TOC-complex core proteins TOC75 and −33 also accumulated to considerably lower levels in the GFP-TOC159GM-K/R:*ppi2* line under low GA presumably to maintain complex stoichiometry (*Figure 3A, B*). Typically, a small fraction of the SUMO target protein pool undergoes SUMOylation under a specific cellular condition (*Johnson, 2004*). The reduced levels under low GA of GFP-TOC159GM-K/R when compared to the wild-type protein were attributed to increased UPS-mediated protein degradation as both proteins recovered to the same protein levels in the presence of the proteasome inhibitor MG132 (*Figure 3D,E*).

We conclude that SUMOylation partially stabilizes TOC159 against UPS-dependent degradation under specific conditions such as low GA during early development. The photosynthesis-associated proteins are considered the preferred transport cargoes of TOC159 because they fail to accumulate in the *ppi2* mutant lacking TOC159. They tended to accumulate to a lesser degree in GFP-TOC159GM-K/R:*ppi2* plants under low GA in the presence of PAC, but not in statistically significant fashion. The explanation may be that the levels of TOC159 are already low in the GFP-TOC159GM-K/R:ppi2 plants in the absence of PAC and allow only for relatively small changes in the accumulation of photosynthesis-associated proteins in the presence of PAC. Nevertheless, the results suggest that SUMOylation at the M-domain of TOC159 serves to fine tune preprotein import under low GA.

Based on the results, we propose a hypothetical model for the role of SUMOylation during early developmental stages, when environmental conditions are unfavorable and GA concentrations are low. As previously demonstrated, the chloroplast import receptor TOC159 is ubiquitylated prior to outer membrane insertion by an unknown E3-ligase other than SP1 and degraded via the UPS. In addition, its G-domain interacts with SUMO3, the physiological consequences of which remain unknown but may regulate interaction with DELLA system. The M-domain may be SUMOylated by SUMO3, protecting to some extent against UPS-dependent degradation. When the environmental conditions become more favorable, GA levels increase, and the GA receptor-DELLA complex is degraded by the UPS and TOC159 liberated for outer membrane insertion. The non-ubiquitinated TOC159 is assembled into the TOC complex thus allowing proplastids to differentiate into chloroplasts (*Figure 4*). In the CHLORAD pathway, cytosolic Cdc48 extracts ubiquitinated TOC proteins from the outer membrane of the chloroplast (*Ling et al., 2019*; *Shanmugabalaji and Kessler, 2019*). The intriguing connection between the Cdc48 and SUMO pathways in chromatin dynamics (*Franz et al., 2016*) suggests that the SUMO pathway might also act on the CHLORAD pathway at specific developmental stage, but we currently do not offer evidence for such a mechanism. Our data specifically implicate SUMOylation and probably SUMO interaction in the control of proplastid to chloroplast transition by regulation of TOC159 levels in early plant development.

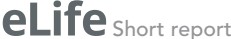

**Figure 4.** Hypothetical model for the SUMOylation-dependent fine-tuning of chloroplast biogenesis at the level of the TOC159 import receptor in early plant development. Environmental conditions influence the concentrations of gibberellic acid (GA) that accumulate upon seed imbibition. When active GA levels are reduced by stress (-GA; left-hand panel), DELLA (RGL2) accumulates and sequesters TOC159, which is then degraded via the UPS. In addition, TOC159 interacts and is also covalently SUMOylated by SUMO3. Covalent SUMOylation protects TOC159 against UPS-mediated degradation and supports the accumulation of photosynthesis-associated proteins in the chloroplast. Any unimported preproteins are degraded in the cytosol via the UPS. When GA concentrations increase (+GA, right-hand panel), the GA receptor complex binds to DELLA, which is degraded by the UPS. TOC159 is then free to assemble into the TOC complex. Protein import is thus enabled, allowing proplastids to differentiate into chloroplasts.

## Materials and methods

### Key resources table

| Reagent type (species) or resource | Designation | Source or reference | Identifiers | Additional information |
|---|---|---|---|---|
| Genetic reagent (*Arabidopsis thaliana*) | pTOC159-GFP-TOC159GM:*ppi2* | *Shanmugabalaji et al., 2018* | | |
| Genetic reagent (*Arabidopsis thaliana*) | pTOC159-GFP-TOC159GM-K/R | This paper | | |

*Continued on next page*

*Continued*

| Reagent type (species) or resource | Designation | Source or reference | Identifiers | Additional information |
|---|---|---|---|---|
| Recombinant DNA reagent | pGBKT7-TOC159G | *Shanmugabalaji et al., 2018* | | Yeast two-hybrid assay |
| Recombinant DNA reagent | pGBKT7-TOC159M | *Shanmugabalaji et al., 2018* | | Yeast two-hybrid assay |
| Recombinant DNA reagent | pGADT7-SUMO1 | This paper | | Yeast two-hybrid assay |
| Recombinant DNA reagent | pGADT7-SUMO2 | This paper | | Yeast two-hybrid assay |
| Recombinant DNA reagent | pGADT7-SUMO3 | This paper | | Yeast two-hybrid assay |
| Recombinant DNA reagent | 35S-GFP-TOC159GM | *Shanmugabalaji et al., 2018* | | In planta CoIP |
| Recombinant DNA reagent | 35S-GFP | This paper | | In planta CoIP |
| Recombinant DNA reagent | 35S-SUMO3-MYC | This paper | | In planta CoIP |
| Recombinant DNA reagent | 35S-GFP-TOC159GM-K/R | This paper | | In planta CoIP |
| Antibody | Anti-TOC75, rabbit polyclonal | *Hiltbrunner et al., 2001* | | 1:1000 |
| Antibody | Anti-TOC33, rabbit polyclonal | *Rahim et al., 2009* | | 1:1000 |
| Antibody | Anti-RGL2, rabbit polyclonal | *Piskurewicz et al., 2008* | | 1:1000 |
| Antibody | Anti-SUMO3, rabbit polyclonal | Agrisera | Cat# AS08349 | 1:1000 |
| Antibody | Anti-UGPase, rabbit polyclonal | Agrisera | Cat# AS05086 RRID:AB_1031827 | 1:5000 |
| Antibody | Anti-PSBA, rabbit polyclonal | Agrisera | Cat# AS05084 RRID:AB_2172617 | 1:10,000 |
| Antibody | Anti-PSBO1, rabbit polyclonal | Agrisera | Cat# AS142824 RRID:AB_1031788 | 1:5000 |
| Antibody | Anti-RBCL, rabbit polyclonal | Agrisera | Cat# AS03037 RRID:AB_2175406 | 1:15,000 |
| Antibody | Anti-RBCS, rabbit polyclonal | Agrisera | Cat# AS07259 RRID:AB_1031806 | 1:5000 |
| Antibody | Anti-PETC, rabbit polyclonal | Agrisera | Cat# AS08330 RRID:AB_2162102 | 1:5000 |
| Antibody | Anti-PSAD, rabbit polyclonal | Agrisera | Cat# AS09461 RRID:AB_1832088 | 1:5000 |
| Antibody | Anti-ATPC, rabbit polyclonal | Agrisera | Cat# AS08312 RRID:AB_2290280 | 1:5000 |
| Antibody | Anti-LHCB2, rabbit polyclonal | Agrisera | Cat# AS01003 RRID:AB_1832080 | 1:10,000 |
| Antibody | Anti-actin, mouse monoclonal antibody | Sigma | Cat# A0480 RRID:AB_476670 | 1:2000 |
| Antibody | Anti-GFP, mouse monoclonal antibody | Takara | Cat#632380 RRID:AB_10013427 | 1:1000 |
| Antibody | Anti-MYC, mouse monoclonal antibody | Cell Signaling | Cat#2276 RRID:AB_331783 | 1:1000 |
| Antibody | Anti-rabbit IgG conjugated with horseradish peroxidase | Millipore | Cat# AP132P RRID:AB_90264 | 1:10,000 |
| Antibody | Anti-mouse IgG conjugated with horseradish peroxidase | Sigma | Cat# A5278 RRID:AB_258232 | 1:10,000 |

*Continued on next page*

| Reagent type (species) or resource | Designation | Source or reference | Identifiers | Additional information |
|---|---|---|---|---|
| Strain, strain background (*Escherichia coli*) | DH5α | Invitrogen | Cat# 18265017 | |
| Strain, strain background (*Agrobacterium tumefaciens*) | C58 | Community resource | | |
| Chemical compound, drug | MG132 | AbMole | Cat# M1902 | |
| Chemical compound, drug | Paclobutrazol | Sigma | Cat# 43900 | |
| Commercial assay or kit | μMACS GFP-tagged beads | Miltenyi Biotech | Cat# 130091125 | |
| Software, algorithm | ImageQuant TL | GE Healthcare | RRID:SCR_014246 | |
| Software, algorithm | GraphPad Prism | Graphpad | RRID:SCR_015807 | |

## Plant materials and growth conditions

The *A. thaliana* Wassilewskija (Ws) ecotype was used as wild types. The *ppi2* mutant and pTOC159-GFP-TOC159GM:*ppi2* line used in this study were in the Wassilewskija (Ws) ecotype as previously described (*Bauer et al., 2000*; *Shanmugabalaji et al., 2018*). Seedlings were grown on MS medium with long-day conditions (16 hr light, 8 hr dark, 120 μmol × m$^{-2}$ × s$^{-1}$, 21°C). Plants were grown on soil either under short-day conditions (16 hr dark, 8 hr light, 120 μmol × m$^{-2}$ × s$^{-1}$, 21°C) for vegetative growth or under long-day conditions (16 hr light, 8 hr dark, 120 μmol × m$^{-2}$ × s$^{-1}$, 21°C) for flower development and seed production.

## Seedling treatment

Surface-sterilized seeds were placed on MS medium supplemented with 2 μM or 5 μM PAC. The plates were incubated under long-day conditions for 3 days. Proteasome inhibitor experiments were performed as described earlier (*Shanmugabalaji et al., 2018*).

## Plant transformation and transgenic lines

The K1370R point mutation was introduced into the binary construct pTOC159-GFP-TOC159GM using a site-directed mutagenesis kit (Agilent-QuikChangeII) with the primers TOC159S3F and TOC159S3R and resulted in pTOC159-GFP-TOC159GM-K/R. The pTOC159-GFP-TOC159GM-K/R construct was introduced into *Agrobacterium tumefaciens* (C58 strain) and stably transformed into heterozygous *ppi2* plants using the floral dip method (*Clough and Bent, 1998*). Transformed plants were selected on phosphinothricin-containing medium, and lines homozygous for the transgene as well as the *ppi2* mutation were isolated and named pTOC159:GFP-TOC159GM-K/R:*ppi2* (referred to as GFP-TOC159GM-K/R:*ppi2* plants).

## Yeast two-hybrid assays

The pGBKT7-TOC159G (BD fusion) and pGBKT7-TOC159M (BD fusion) vector were introduced into the yeast strain Y2H GOLD as previously described (*Shanmugabalaji et al., 2018*). The full-length cDNA sequences of SUMO1, SUMO2, and SUMO3 were amplified using primers (S1F, S1R, S2F, S2R, S3F, S3R), digested with NdeI/EcoRI, and ligated into the corresponding sites of the pGADT7 vector. The empty bait vector (BD) was used as a negative control. Co-transformants were selected on SD –Leu –Trp and SD –Leu –Trp –His plates.

## Chlorophyll measurements

Chlorophyll levels were measured from four biological replicates as previously described (*Agne et al., 2009*).

## Confocal laser scanning microscopy

One week old seedlings of the GFP-TOC159GM:*ppi2* and GFP-TOC159GM-K/R:*ppi2* lines were directly observed under a TCS SP5 II confocal laser scanning microscope (Leica Microsystems, Heerbrugg, Switzerland). Samples were excited by a 488 nm argon laser and detected with a 524–546 nm band-pass filter for GFP and chlorophyll autofluorescence seen in the 650–720 nm range with ×60. The digital images were acquired using LAS AF digital system (version: 2.0.0 build 1934, Leica Microsystems, Wetzlar, Germany). The LCS lite software (Leica) was used for the analysis of the picture.

## Protein extraction and immunoblotting

Identical amounts of samples were collected, and proteins were extracted using AP extraction buffer (100 mM Tris pH 8, 2% b-mercaptoethanol, 4% sodium dodecyl sulfate, 20% glycerol) followed by acetone precipitation (*Piskurewicz and Lopez-Molina, 2011*). The sodium dodecyl sulfate–polyacrylamide gel electrophoresis and immunoblotting were performed according to the standard protocols. To probe the blots, primary antibodies recognizing TOC75 (*Hiltbrunner et al., 2001*), TOC33 (*Rahim et al., 2009*), GFP (Takara), SUMO3 (Agrisera), MYC (Cell signling), UGPase (Agrisera), actin (Sigma), and RGL2 (*Piskurewicz et al., 2008*) were used. As markers for photosynthesis-associated proteins, antibodies recognizing PSBA, PSBO1, RBCL, RBCS, PETC, PSAD, ATPC, and LHCB2 were purchased from Agrisera. Secondary antibodies were anti-rabbit IgG conjugated with horseradish peroxidase (Millipore) or goat anti-mouse IgG conjugated with horseradish peroxidase (Sigma). Chemiluminescence was detected using ECL Plus Western Blotting Detection Reagents (Pierce) and developed using a GE Amersham Imager 600. Band intensities were quantified using ImageQuant TL (GE Healthcare) software.

## In planta co-immunoprecipitation (CoIP) from transient expression system in *N. benthamiana*

Full-length SUMO3 was PCR amplified from cDNA using the primers SUMO3–F(GATE) and SUMO3–R2(GATE) and inserted into the pENTR221 vector by BP clonase (Invitrogen). It was recombined into the pEarleyGate 203 binary vector with LR clonase to obtain a 35S-SUMO3-MYC fusion construct. The binary vector 35S-GFP-TOC159GM used in this study has been described previously (*Shanmugabalaji et al., 2018*). The point mutation was introduced in the binary construct 35S-GFP-TOC159GM by using a site-directed mutagenesis kit (Agilent-QuikChangeII) with the primers TOC159S3F and TOC159S3R, as results we obtained 35S-GFP-TOC159GM-K/R. The 35S-GFP, 35S-SUMO3-MYC, 35S-GFP-TOC159GM, and 35S-GFP-TOC159GM-K/R were introduced into *A. tumefaciens* (C58 strain). And co-infiltrated into 3 week old *N. benthamiana*. Immunoprecipitation to isolate the protein complexes from total protein extracts using GFP-tagged microbeads (Miltenyi Biotec) has been described previously (*Shanmugabalaji et al., 2018*). Anti-GFP antibody was used to detect GFP-TOC159, GFP-TOC159GM-K/R. Anti-SUMO3 and anti-MYC antibodies were used to identify the SUMOylated TOC159GM.

## Quantification and statistical analysis

For protein quantification on immunoblots, ImageQuant TL (GE Healthcare) software was used to measure band intensities, and the data are shown as mean ± SEM. The statistical analysis (two-tailed t test) was performed in GraphPad Prism 8.4.3 (http://www.graphpad.com), with p values higher than 0.05 being considered non-significant (n.s.), while p values lower than 0.05 being considered significant for the analyzed data and indicated as follows: *p<0.05; **p<0.01; ***p<0.005.

## Acknowledgements

This work was supported by grants from the Swiss National Science Foundation (31003A_156998 and 31003A _176191) and by the University of Neuchâtel.

## Additional information

### Funding

| Funder | Grant reference number | Author |
|---|---|---|
| Schweizerischer Nationalfonds zur Förderung der Wissenschaftlichen Forschung | 31003A_156998 | Felix Kessler |
| Schweizerischer Nationalfonds zur Förderung der Wissenschaftlichen Forschung | 31003A _176191 | Felix Kessler |

The funders had no role in study design, data collection and interpretation, or the decision to submit the work for publication.

### Author contributions

Sonia Accossato, Data curation, Validation, Investigation, Methodology, Writing - original draft; Felix Kessler, Supervision, Funding acquisition, Validation, Writing - original draft, Project administration, Writing - review and editing; Venkatasalam Shanmugabalaji, Conceptualization, Data curation, Formal analysis, Supervision, Validation, Investigation, Visualization, Writing - original draft, Project administration, Writing - review and editing

### Author ORCIDs

Felix Kessler https://orcid.org/0000-0001-6409-5043
Venkatasalam Shanmugabalaji https://orcid.org/0000-0002-3855-6958

### Decision letter and Author response

Decision letter https://doi.org/10.7554/eLife.60968.sa1
Author response https://doi.org/10.7554/eLife.60968.sa2

## Additional files

### Supplementary files

• Supplementary file 1. List of primers used in this study.

• Transparent reporting form

### Data availability

All data generated or analysed during this study are included in the manuscript and supporting files. Source data files have been provided for Figures 2 and 3.

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
