## [Decision Letter]

**Acceptance summary:**

The authors provide evidence that chloroplast protein import machinery can be SUMOylated and propose a model in which interplay between SUMOylation and ubiquitination of a chloroplast protein import receptor could influence proplastid development into chloroplasts in germinating seeds.

**Decision letter after peer review:**

Thank you for submitting your article "SUMOylation contributes to proteostasis of the chloroplast protein import receptor TOC159 during early development" for consideration by *eLife*. Your article has been reviewed by three peer reviewers, one of whom is a member of our Board of Reviewing Editors, and the evaluation has been overseen by Christian Hardtke as the Senior Editor. The following individual involved in review of your submission has agreed to reveal their identity: Danny Schnell (Reviewer #2).

The reviewers have discussed the reviews with one another and the Reviewing Editor has drafted this decision to help you prepare a revised submission.

Summary:

Based on previous results that Toc159 can be modified by SUMOylation by SUMO3, the authors confirm interactions between a truncated version of TOC159 and SUMO3 using Y2H and co-IPs. They find that this interaction is substantially decreased with a K1370->R mutant that abolishes a high-confidence SUMOylation site, but that this K1370R mutant can complement the *ppi2* (toc159) mutant as efficiently as the wild type version of TOC159. However, under conditions that normally deplete the TOC complex (paclobutrazol-induced low GA conditions during germination), they find that the K1370R mutant is more sensitive to degradation, but ultimately this had no significant effect on chloroplast protein import. The authors propose a model in which under low GA conditions, TOC159 is either SUMOylated and stable or ubiquitinated and degraded, suggesting an important balance between these post-translational modifications. All three reviewers agree that while the findings are interesting, additional experimentation, particularly controls, are required to support the model that the authors present.

Essential revisions:

1) Although compelling, the authors' hypothesis would be strengthened if they could demonstrate the impact of SUMOylation on the accumulation of photosynthetic proteins or chloroplast biogenesis. Can the authors test additional photosynthetic proteins to see if they can detect a defect in accumulation? Did the authors consider including stress conditions to test if SUMOylation impacts chloroplast biogenesis or protein accumulation under non-ideal conditions? Alternatively, does the TOC159GM-K/R impact de-etiolation?

2) The model speculates that SUMOylated TOC159 is not incorporated into the chloroplast membrane or into a functional TOC complex and that "deSUMOylation of TOC159 by a SUMO protease (ULP) assists assembly into the TOC complex". What is the evidence that SUMO-Toc159 is cytoplasmic and not at the chloroplast envelope? Figure 2D shows all Toc159 at the chloroplast envelope. Is it possible that SUMOylation works downstream of DELLA to stabilize TOC complexes that are assembled at the membrane? Please present these data or revise the model and associated text accordingly.

3) It is a stretch to call the TOC159GM-K/R mutant "non-SUMOylatable", since there are there are clear bands on the K/R lanes of Figure 1F. There are several other high confidence SUMOylation sites in TOC159. What was the rationale for selecting only one SUMOylation site on Toc159? Please present a blot with TOC159GM-K/R and SUMO3-Myc alone in Figure 1F so that readers can evaluate the anti-SUMO/anti-MYC signal in this case and revise the text accordingly unless all anti-SUMO signal is abolished in this experiment.

4) Controls seem to be missing from the co-IPs in Figure 1. Please present a GFP/MYC-only (or even better, SUMO1-MYC) as a negative control.

5) The authors should switch the vectors of the yeast two-hybrid constructs (Figure 1) to demonstrate that the interaction is not a vector artifact. This is a standard control for this assay.

6) The authors should quantify the protein bands in Figure 3C. Although they conclude the RGL2 does not change, the levels appear to be lower in the K/R mutant under normal and PAC treatment. Although this is beyond the scope of the current study, it would be interesting to determine if the K/R plants exhibit a germination defect if RGL2 levels are lower.

7) In Figure 3A to show clear protein stability effects for K/R and WT TOC159 in the during PAC treatment the authors would really need to use cycloheximide treatment to verify the stability of the K/R form to substantiate their claims.

8) What are the possible effects of using a truncated form of TOC159? This could also explain why they see a knock-on reduction in TOC75 and TOC33. The A-domain is protease sensitive, but could this sensitivity be underpinned by post-translational modifications? Please provide additional experimental evidence or revise the discussion of these results.

9) The "phylogenetic analysis" to document that K1370 is conserved need to be rigorously conducted and methods need to be reported to give context to the "alignment" presented in Figure 1E.

---

## [Author Response]

Essential revisions:1) Although compelling, the authors' hypothesis would be strengthened if they could demonstrate the impact of SUMOylation on the accumulation of photosynthetic proteins or chloroplast biogenesis. Can the authors test additional photosynthetic proteins to see if they can detect a defect in accumulation? Did the authors consider including stress conditions to test if SUMOylation impacts chloroplast biogenesis or protein accumulation under non-ideal conditions? Alternatively, does the TOC159GM-K/R impact de-etiolation?

We have included more photosynthetic proteins in Figure 3—figure supplement 5 Western blot. The amount of photosynthetic protein accumulation is statistically insignificant in TOC159GM-K/R:*ppi2* compared to TOC159GM:*ppi2* under low gibberellic acid (GA). The new results confirm, however, that photosynthetic proteins trend lower in TOC159GM-K/R.

We tested whether de-etiolation impacts TOC159GM-K/R but did not detect any difference in the rate of greening after germination in the dark when compared to WT and TOC159GM:*ppi2* (Author response image 1).

It will be interesting to test whether stress conditions impact chloroplast biogenesis or protein accumulation in the future while the main goal of this manuscript was to reveal the effects of SUMOylation on TOC159 accumulation in the early developmental stage.

**Author response image 1. sa2fig1:** A de-etiolation assay was carried out with WT, GFP-TOC159GM:*ppi2* and GFP-TOC159GM-K/R:*ppi2* plant lines (three independent biological replicates with at least 50 seedlings). The seeds were stratified and exposed for 3h to 120 µmol x m-2 x s^-1^ of white light, then grown in the dark for 6 days. The etiolated seedlings were transferred to continuous light for two more days. The survival rate of de-etiolated seedlings based on cotyledon greening. Error bars indicate ± SEM (n = 3).

2) The model speculates that SUMOylated TOC159 is not incorporated into the chloroplast membrane or into a functional TOC complex and that "deSUMOylation of TOC159 by a SUMO protease (ULP) assists assembly into the TOC complex". What is the evidence that SUMO-Toc159 is cytoplasmic and not at the chloroplast envelope? Figure 2D shows all Toc159 at the chloroplast envelope. Is it possible that SUMOylation works downstream of DELLA to stabilize TOC complexes that are assembled at the membrane? Please present these data or revise the model and associated text accordingly.

Figure 2D shows that TOC159GM and TOC159GM-K/R localize to the chloroplast envelope in developed green seedlings (optimal gibberellic acid (GA) conditions). Under low GA, TOC159 interacts with RGL2 in the nucleo-cytoplasmic compartment and is degraded prior to the insertion into the chloroplast outer membrane (Shanmugabalaji et al., 2018). However, we do not have evidence supporting that "deSUMOylation of TOC159 by a SUMO protease (ULP) assists assembly into the TOC complex". Therefore we revised the model and the text.

3) It is a stretch to call the TOC159GM-K/R mutant "non-SUMOylatable", since there are there are clear bands on the K/R lanes of Figure 1F. There are several other high confidence SUMOylation sites in TOC159. What was the rationale for selecting only one SUMOylation site on Toc159? Please present a blot with TOC159GM-K/R and SUMO3-Myc alone in Figure 1F so that readers can evaluate the anti-SUMO/anti-MYC signal in this case and revise the text accordingly unless all anti-SUMO signal is abolished in this experiment.

Some of the predicted SUMOylation sites are present in the N-terminal A-domain. We have previously demonstrated that constructs lacking the A-domain and consisting only of the G- and M-domains (TOC159GM) and a tag at the N-terminus complemented the Toc159 null mutant *ppi2 (*Agne et al., 2009) and were therefore functional. Because the A-domain of TOC159 is unstable (i.e. resulting in multiple TOC159 bands due to fragmentation) it is normally necessary to work with N-terminally tagged TOC159 constructs without the A-domain (this reply is also valid for comment 8).

We analysed the TOC159GM domain for SUMO sites using the GPS SUMO prediction algorithm with a high threshold (http://sumosp.biocuckoo.org/online.php). We found two predicted SUMOylation sites at the M-domain but not in the G-domain (Figure 1—figure supplement 2A). Based on the superior SUMOylation consensus sequence pattern, p-value and score, we selected the site at 1370. The in planta SUMOylation assay demonstrates that the vast majority of SUMOylation of TOC159GM occurs at TOC159GM-K1370. (Figure 1F and new Figure 1—figure supplement 2B). We therefore concluded that any other potential sites are not SUMOylated efficiently or that the observed signals were spurious.

We didn't include the SUMO3-Myc alone in the experiments. TOC159GM alone (Figure 1F, lane 1) and TOC159GM-K/R alone (Figure 1F, lane 2) did not result in any high molecular weight anti-MYC signals in the Western blot.

4) Controls seem to be missing from the co-IPs in Figure 1. Please present a GFP/MYC-only (or even better, SUMO1-MYC) as a negative control.

As a control we have added a new in planta SUMOylation experiment showing co-infiltration of GFP with SUMO3-MYC ( lane 1), GFP-TOC159GM with SUMO3-MYC ( lane 2) and GFP-TOC159GM-K/R and with SUMO3-MYC ( lane 3). (Figure 1—figure supplement 2B). The co-immunoprecipitated GFP with SUMO3-MYC ( lane 1), did not result in any high molecular weight signal with anti-MYC compared with GFP-TOC159GM with SUMO3-MYC sample ( lane 2). The results further confirm the specificity of SUMOylation of GFP-TOC159GM.

5) The authors should switch the vectors of the yeast two-hybrid constructs (Figure 1) to demonstrate that the interaction is not a vector artifact. This is a standard control for this assay.

While we have not been able to do this in useful time due to the pandemic (limited lab time), we added a new Figure 1C. It shows an alternative experiment (with available reagents) to confirm the SUMO3 binding to TOC159 using co-immunoprecipitation after transient expression of TOC159GM together with SUMO3-Myc in *N. benthamiana*.

6) The authors should quantify the protein bands in Figure 3C. Although they conclude the RGL2 does not change, the levels appear to be lower in the K/R mutant under normal and PAC treatment. Although this is beyond the scope of the current study, it would be interesting to determine if the K/R plants exhibit a germination defect if RGL2 levels are lower.

We have quantified the RGL2 protein accumulation from three independent experiments and included it in new Figure 3—figure supplement 3A and B. In conclusion, RGL2 protein accumulation in GFP-TOC159GM and GFP-TOC159GM-K/R lines are almost the same under low GA conditions.

Also, we didn't observe any seed germination phenotype in GFP-TOC159GM and GFP-TOC159GM-K/R lines (not shown).

7) In Figure 3A to show clear protein stability effects for K/R and WT TOC159 in the during PAC treatment the authors would really need to use cycloheximide treatment to verify the stability of the K/R form to substantiate their claims.

We were not quite sure what the reviewer meant here. But we carried out the following experiment: GFP-TOC159GM and GFP-TOC159GM-K/R lines grown on PAC (low GA) (Author response image 2, lane 1, 2) were further treated with cycloheximide (100µM) for 3 hours (Author response image 2, lane 3, 4). The results revealed no effect of cycloheximide on either TOC159GM-K/R or TOC159GM levels (Author response image 2). This would suggest that any differences observed in protein abundance were not somehow due to de novo protein synthesis but to proteolysis.

**Author response image 2. sa2fig2:** The TOC159 protein accumulation from GFP-TOC159GM:ppi2and GFP-TOC159GM-K/R:ppi2under low GA conditions and further treated with cycloheximide. (**A**) Total protein extracts of three days old GFP-TOC159GM:*ppi2* or GFP-TOC159GM-K/R:*ppi2* grown seedlings grown on PAC and subsequently treated with or without cycloheximide were analyzed by immunoblotting using anti-GFP antibodies and Ponceau staining for a loading control.(**B**) The specific bands corresponding to GFP were quantified from two independent experiments (A). The quantified bands were normalized to GFP-TOC159GM in GFP-TOC159GM:*ppi2* without cycloheximide. Error bars indicate ± SEM (n = 2).

8) What are the possible effects of using a truncated form of TOC159? This could also explain why they see a knock-on reduction in TOC75 and TOC33. The A-domain is protease sensitive, but could this sensitivity be underpinned by post-translational modifications? Please provide additional experimental evidence or revise the discussion of these results.

See reply to question 3 regarding the use of a truncated form of TOC159 lacking the A-domain. In a previous report we demonstrated that TOC159 A domain cleavage, occurs at a conserved protease cleavage site (Agne et al., 2010) and at a specific ratio of cleaved to uncleaved TOC159 (Zufferey et al., 2017). But how this protease sensitivity is underpinned by post-translational modifications is unknown.

Regarding the knock-on reduction of TOC75 and TOC33: we previously demonstrated that under low GA, the full length TOC159 protein level is reduced by UPS implicating an unidentified E3-ligase, whereas the reduced levels of TOC75 and TOC33 implicate the outer membrane E3 ligase SP1 (Shanmugabalaji et al., 2018). In this manuscript, we attribute the knock-on reductions of TOC75 and TOC33 protein to the same mechanism.

9) The "phylogenetic analysis" to document that K1370 is conserved need to be rigorously conducted and methods need to be reported to give context to the "alignment" presented in Figure 1E.

Thank you for pointing out this requirement regarding the "phylogenetic analysis". We used CLUSTAL Omega (1.2.4) multiple sequence alignment. We have now included this in Figure 1—source data 1 and mentioned it in the figure legend.